# The Change in the Structure and Functionality of Ferritin during the Production of Pea Seed Milk

**DOI:** 10.3390/foods11040557

**Published:** 2022-02-16

**Authors:** Yilin Xing, Jiaqi Ma, Qimeng Yao, Xuemin Chen, Jiachen Zang, Guanghua Zhao

**Affiliations:** Beijing Key Laboratory of Functional Food from Plant Resources, College of Food Science and Nutritional Engineering, China Agricultural University, Beijing 100083, China; viestana@163.com (Y.X.); majiaqi971026@163.com (J.M.); QM_YAO@163.com (Q.Y.); charmainexm@163.com (X.C.); zangjiachen@cau.edu.cn (J.Z.)

**Keywords:** ferritin, pea seed milk, thermal processing, structure

## Abstract

Understanding the effect of thermal treatment on the physical and chemical properties of protein and its mechanisms has important theoretical implications in food science. Pea seed ferritin (PSF) is an iron storage protein naturally occurring in pea seeds, which represents a promising iron supplement. However, how thermal processing affects the structure and function of PSF remains unknown. In this work, during the production of pea seed milk, we investigated the effect of thermal treatments at boiling temperature for two different times (5 and 10 min), respectively, on the structure and function of PSF. The results demonstrated that thermal treatment resulted in a pronounced change in the primary, secondary, and tertiary structure, iron content, and iron oxidation activity of PSF. However, the shell-like structure of PSF can be kept during the processing of pea seed milk. Interestingly, upon thermal treatment, both thermal-treated samples exhibit larger higher iron absorption rate by Caco-2 than untreated PSF at the same protein concentration. Such an investigation provides a better understanding of the relationship between the structure and function of food protein, as affected by thermal treatment.

## 1. Introduction

Iron is a necessary micronutrient that acts as a catalytic center for a wide variety of metabolic functions, such as oxygen transport, cell respiration, energy production, enzyme activating, and gene regulation [1,2,3]. The human body, weighing 70 kg, contains 4 to 5 g of iron, approximately. However, iron deficiency has been reported to affect about two billion individuals, including a great number of pregnant women and young children [4]. It is also a major cause of anemia worldwide [5] and is one of the global issues for disability and even death [6]. Despite the pharmaceutical iron uptake, which may lead to adverse effects, such as diarrhea, constipation, and decreased growth [7,8], the dietary intake of iron element is a friendlier way for daily iron supplement. Nutritional iron is usually divided into two types: heme, which is majorly stored in meat products; and non-heme, with broader plant-originated sources but lower bioavailability [9,10]. On the other hand, meat consumption is still unavailable for the economically underdeveloped areas and thus a great amount of people cannot eat meat for physical or religious reasons. In addition, in terms of the global pressure on the demand for energy, utilization of plant-based food instead of animal-based food appears to be a more sustainable strategy [11]. Thus, developing the bioavailability of non-heme iron from plant resources for human absorption is of great importance. Recently, ferritin from legume seeds has drawn a lot of attention as a novel way for iron supplementation [12,13], because over 90% of the total iron is stored in the ferritin nanocage [14,15].

Ferritin, a very large, unique, and conserved protein responsible for iron storage and detoxification, is encoded in the genomes of widely distributed species, from Archaea to humans, and expressed in most cell types [16,17]. The ferritin is endowed with a shell-like structure which is usually composed of 24 subunits that self-assemble into a *F432* symmetrical manner, and its shell-like structure is stabilized by four types of subunit-subunit interfaces, *C*_2_, *C*_3_, *C*_4_, and *C*_3_*C*_4_ interfaces. The outer and inner diameters of ferritin are approximately ~12 nm and ~8 nm, respectively. Remarkably, one ferritin molecule can carry approximately up to ~4500 iron atoms, and this is why ferritin is regarded as a promising iron supplement [18]. The mature plant ferritin is commonly constituted of two different H-subunits, H (heavy)-1 and H-2, respectively. The H-subunit contains a ferroxidase center, which is responsible for fast iron oxidation. What is more, there is a unique domain of extension peptide (EP) at the N-terminal of each subunit of plant ferritin surrounding the cage-like surface, as shown in Figure 1A. Our research groups have found that the EP domain plays a critical role in the self-degradation and aggregation of plant ferritin [19,20]. It also has been proven that ferritins can pass through the gastrointestinal tract and can be absorbed in the form of an intact ferritin-iron complex with a higher iron bioavailability [21,22].

It is very meaningful to investigate the change in the structure and function of ferritin during food processing, as protein is very sensitive to thermal treatment, and such a change will have a marked effect on the iron supplementation of ferritin. Recently, Masuda has proven that after an 80 °C treatment, ferritin extracted from soybean tofu exhibits a gradient decrease tendency [23]. Our research group demonstrated that thermal treatment at 60–80 °C may greatly improve the pea seed ferritin’s storage stability and monodispersity [24]. In addition, our study has demonstrated that the EP domain of soybean seed ferritin makes great contribution to its high thermostability [25]. More recently, Li and co-workers have studied the structure and function of oyster ferritin [26,27].

Generally, these studies mostly focused on the change of pure ferritin upon various thermal treatments. There has been little information on the structure and function of ferritin occurring in a complex food system. As one of the most widely consumed legume seeds, pea (*Pisum sativum*) is regarded as a potential food source with its high nutritional value and low cost [28]. The kinds and amounts of pea product have also increased prominently in recent decades, and include pea tofu, pea cake, pea meal, and pea seed milk. Ferritin is the most important iron-storing protein in legume seeds, including soybeans and pea seeds. In this study, we firstly made pea seed milk, a popular beverage in China, and subsequently determined the effect of pea seed milk production at boiling temperature on the structure and function of pea seed ferritin (PSF). The results demonstrated that thermal treatment greatly affected the structure and function of PSF.

## 2. Materials and Methods

### 2.1. Materials and Chemicals

Pea (*Pisum sativum*) seeds were purchased from the local market (Beijing, China). Sodium lauryl sulfate (SDS), tetramethylethylenediamine (TEMED), polyvinyl pyrrolidone (PVP), tris hydroxymethyl methyl aminomethane (Tris), 3-morpholinopropanesulfonic acid (Mops), the BCA protein quantitation kit, and the dialysis membrane were purchased from Solarbio Co. (Beijing, China). All reagents used in this study were analytical grade and all solutions were prepared with ultrapure water.

### 2.2. Production of Pea Seed Milk and Purification of Ferritin

The production of pea seed milk and purification of ferritin were carried out according to the following steps with some modifications [24]. The pea seeds were firstly soaked and kept in distilled water for 12 h. Then, the pea seed milk was obtained by blending the soaked pea seeds with 50 mM Tris-HCl buffer (pH = 7.5, 1% PVP) and homogenizing the mixture at 12.0 krpm for 2 min, three times. Compared to the samples without thermal treatment, the pea seed milk was further heated to 100 °C and maintained at boiling state for a different time. The boiled pea seed milk was treated with an ice bath after the thermal treatment to control the exact heating time, as described. After filtering out the residue with a 200-mesh strainer, the solution was centrifuged at 5000× *g* for 10 min, with the supernatant collected. The amounts of 0.2 M MgCl_2_ and 0.3 M sodium citrate are added into the supernatant to salt out the protein in the supernatant, with stirring for 1 h and further standing for 8 h. Next, the solution was centrifuged at 10,000× *g* for 30 min. The obtained brown-colored pellet was dissolved in 50 mM Tris-HCl buffer (pH = 7.5), and was dialyzed against the same buffer 3 times for 18 h with a dialysis membrane (100 kDa MWCO, 18 mm width).

The ferritin was purified by ion exchange chromatography in a DEAE-Cellulose column (height: 60 cm, diameter: 1 cm, particle size 50 μm), with a gradient of NaCl (0–1.0 M) in 50 mM Tris-HCl buffer (pH = 7.5) at a flow rate of 4.0 mL/min, and the fraction was collected and performed in SDS-PAGE (polyacrylamide gel electrophoresis) and Native-PAGE, to determine the purity of PmFer. The high-purity fraction was further purified by a Sephacryl S-300 molecular sieve column (height: 150 cm, diameter: 1.8 cm, particle size 25–75 μm) with 50 mM Tris-HCl buffer (pH = 7.5) containing 0.15 M NaCl, at the rate of 2.0 mL/min. The purified ferritin solution was concentrated by a 100 kDa ultrafiltration centrifuge tube. All operations above were performed at 4 °C, unless further directed, to prevent protein denaturation or degradation. The protein concentrations of ferritin samples were determined according to the BCA protein quantitation kit.

### 2.3. Electrophoresis

SDS-PAGE experiments were conducted with 15% polyacrylamide gels. Mixed with equal volume of 2 × loading buffer, followed by a 5-min heat treatment in boiling water, 20 µL protein sample (~10 μg protein contained) was loaded into each well. Electrophoresis was performed under constant voltage conditions at 150 V. Native-PAGE electrophoresis was performed with a 5–20% gradient gel, with the samples untreated by boiling water and mixed with a loading buffer, under constant voltage conditions at 180 V and 4 °C. After electrophoresis, the gels were stained with Coomassie brilliant blue R-250.

SDS-PAGE electrophoresis marker is composed of: phosphorylase B: 97.4 kDa, bovine serum albumin: 66.2 kDa, rabbit actin: 43 kDa, bovine carbonic anhydrase: 31 kDa, pancreatin inhibitor: 20.1 kDa, and lysozyme: 14.4 kDa; the Native-PAGE electrophoresis marker is: thyroglobulin: 669 kDa, horse spleen ferritin: 440 kDa, catalase: 232 kDa, lactate dehydrogenase: 140 kDa, and bovine serum albumin: 66 kDa [29].

### 2.4. Transmission Electron Microscopy (TEM)

The ferritin sample was dialyzed against 50 mM Mops buffer (pH 7.0) for 6 h three times at 4 °C with the same type of membrane as above, followed by dilution with the same buffer to a concentration of 1 μM, and then placed (10 μL) on carbon-coated copper girds. The solution was kept on the mesh for 5 min. After the extra solution removed by filter paper, the PmFer sample was negatively strained with 2% uranyl acetate solution for 5 min. Then, the excess dying solution was removed again, and the remaining samples were left until completely dry for TEM micrograph. In this experiment, the TEM micrographs were obtained by a Hitachi H-7650 transmission electron microscope at the voltage of 80 kV.

### 2.5. Dynamic Light Scattering (DLS)

In the experiment, a Viscotek 802 dynamic light-scattering instrument was used to perform DLS analysis on PSF samples (~1 μM) without heat treatment and under different heating time conditions (5 min and 10 min). Before DLS measurement, the ferritin samples were centrifuged at 10,000× *g*, 10 min to remove bubbles or large particles that would effect the results. In addition, the reaction cell was cleaned with ultrapure water repeatedly, and the reaction system was stabilized for 2 min before the determination. Under the measuring temperature (25 °C), the hydration radius of the ferritin samples (1 μM) was calculated by dynamic light-scattering (DLS), by measuring each sample 10 times. The results obtained were used to analyze the hydration radius (R_H_) of the protein by Origin 8.5 software.

### 2.6. Demineralization

Iron ions were removed from ferritin samples by dialyzing against 1 L each of 20 mM Tris-HCl buffer (pH 7.5, 200 mM NaCl, 3 mM EDTA, 1:400 ammonium thioglycolate) 4 times, while reducing and chelating the iron, followed by four dialysis steps against 1 L each of 20 mM Tris-HCl buffer (pH 7.5, 200 mM NaCl) to remove other ions in the former buffer [30].

### 2.7. Fluorescence Spectra

The fluorescence measurements of PmFer samples were carried out by a Cary Eclipse spectrofluorometer (Varian), with the width of excitation and emission slit both set as 5 nm and the excitation wavelength set as 280 nm. The emission wavelength of ferritin was detected from 290 nm to 500 nm at 25 °C and the spectra were recorded as an average of three scans.

### 2.8. Circular Dichroism (CD) Spectra

The ferritin samples (1.0 μM) were prepared and transferred into a quartz cuvette with a 1.0 cm path length and a 1.0 nm bandwidth. The data of the circular dichroism spectra were obtained by a Pistar π-180 spectrometer (Applied Photophysics, UK) under the scanning rate of 50 nm/min. The spectra were recorded in the range of 180–260 nm, as an average of three scans with the 50 mM Mops solution background subtracted. The percentage of secondary structure, including the α-helix, β-turn, β-sheet, and random coil, was calculated by using CDNN software.

### 2.9. Inductively Coupled Plasma Mass Spectrometry (ICP-MS)

The ICP-MS was used to determine the iron content of the ferritin samples. A total of 200 μL of PmFer solution were taken from each sample, then added with 1 mL of nitric acid to digest for 20 min, and diluted to 20 mL with ultrapure water. The working curve was plotted with a standard solution containing iron (commercially available).

### 2.10. Ferrous Iron (II) Rapid Oxidation Kinetics

The determination of the kinetics of rapid oxidation of ferrous iron was conducted with a Varian Cary 50 spectrophotometer and fresh FeSO_4_ solution (10 mM) in 10 mM HCl to avoid instant oxidation of Fe(II) to Fe(III). A total of 1.0 mL apo ferritin (0.5 mM), from each group, respectively, was mixed with 20 μL FeSO_4_ (pH = 2.0, 10 mM) at 25 °C. The light absorbance at 305 nm was measured for the determination of μ-oxo diFe^3+^ species, using the software Origin 8.5 to plot the kinetics curve of the rapid oxidation.

### 2.11. Cell Absorption

A 0.5 mL suspension of Caco-2 cells was seeded into the apical compartment of 12-well transwell culture plate in DMEM complete medium at a density of about 1 × 10^5^ cells/mL, with 1.5 mL DMEM in each basal compartment. The cells were incubated at 37 °C, 5% CO_2_–95% air atmosphere and 95% relative humidity, and the culture medium was changed every alternate day to maintain the cells in the logarithmic growth. The transepithelial electrical resistance (TEER) was measured with a Millipore cell resistance meter each day until the monolayer became confluent, which means reaching a resistance of above 500 Ω·cm^−2^.

After 6–7 days, the monolayer model was established. The medium of compartments was replaced by DMEM (2% FBS), and the plate was incubated for 30 min. Then the medium was removed, with 0.5 mL of PSFm (~1.0 μM) added in the donor compartment and 1.5 mL of Hank’s balanced salt solution (HBSS) in the acceptor compartment. After 1.5 h the apical compartments were washed by 1 mL PBS, mixed with the medium in the basal part and stored at −20 °C. The iron content was later determined by ICP-MS.
(1)$$\mathrm{Iron}\;\mathrm{absorption}\;\mathrm{rate}\;(\%)=\frac{\mathrm{the}\;\mathrm{amount}\;\mathrm{of}\;\mathrm{iron}\;\mathrm{absorbed}\;\mathrm{into}\;\mathrm{the}\;\mathrm{acceptor}\;\mathrm{compartment}}{\mathrm{the}\;\mathrm{total}\;\mathrm{iron}\;\mathrm{amount}}\;\times \;100$$

## 3. Results and Discussion

### 3.1. The Change of the Structure of PSFm during the Production of Pea Seed Milk

In former research, the crystal structure of soybean ferritin was already solved and reported [31]. Even though there is no reported structure of PSF, it shares high sequency similarity with soybean ferritin, as shown in Figure 1B. We predicted the 3D structure of PSF by the transform-restrained Rosseta [32] and aligned it with the soybean ferritin. The superposition of the two structures coincided with each other with low RMSD (Figure 1C). The one subunit of PSF is composed of four long α-helices (A-helix, B-helix, C-helix, and D-helix), a short E helix, loop between each helix domain, and an EP domain.

As demonstrated in the results of the SDS-PAGE analysis of Figure 2A, there are clear bands around 28 kD in the samples, which correspond to the PSFm subunit. It appears that PSFm molecules still maintain its water solubility upon 100 °C boiling thermal treatments at both 5 and 10 min, respectively, suggesting that PSF molecules are very stable, and can be resistant against thermal treatments at boiling temperature. In contrast, when pure PSF molecules were used, most of them were denatured upon thermal treatment at above 90 °C [24]. This difference in thermal stability between pure PSF and PSF in pea seed milk most likely stemmed from their different environments. We believe that other components, such as starch, fiber, and other proteins (globulin and albumin) in pea seeds [33], could protect PSF from thermal-induced damage.

To obtain more information on the structure of PSFm with two different boiling times (5 and 10 min), we purified PSFm from pea seed milk according to our reported method [24]. SDS-PAGE analyses revealed that the molecular weight (MW) of thermal-treated ferritins decreased to a little extent as compared to PSF from untreated pea seeds, suggesting that the primary structure of PSFm was slightly damaged. Subsequently, we determined whether the secondary structure of PSFm was altered during the production of pea seed milk by CD spectroscopy. As shown in Figure 2B, PSF from untreated pea seeds has an obvious negative ellipticity in the far-UV spectrum at 208 and 222 nm, being in good agreement with previous results demonstrating that the secondary structure of pea seed ferritin is rich in α-helix [28]. The curve fitting of untreated ferritin with program K2D2 produced a content of 65% α-helix, 13% β-turn, and 21% random coil. In contrast, there is a shift of the negative ellipticity to around 225 nm with PSFm upon thermal treatment for 5 min, leading to the different content of 53% α-helix, 16% β-turn, and 29% random coil. With an increase in boiling time from 5 min to 10 min, the content of α-helix of PSFm decreased to 51%, while both β-turn and random coil increased to 16% and 32%, respectively. These results indicated that the secondary structure of PSF was denatured to some extent during thermal processing, and about 10% of α-helix was transformed into random coils due to thermal processing. The EP domain, which is located on the exterior surface of plant ferritin, is sensitive to heating. The crystal structure of thermal-treated PSFm exhibited no density in EP domain, indicating that the rigid structure of this part was destroyed by high temperature [25]. During thermal treatment, the denaturation of the EP domain will lead to a decrease in α-helix and an increase in the random coil of the PSFm in total, which can explain the change of the secondary change of PSFm. Except for the EP domain, it is also possible that the four-helix bundle of each PSF subunit may also be denatured to some extent during the thermal processing, losing its rigid structure.

Except for the secondary structure, the tertiary structure of PSFm was also monitored during the production of pea seed milk by fluorescence spectrophotometry, which is sensitive to the microenvironment surrounding the fluorophore residues. According to Figure 2C, there is a unique peak of PSF purified from untreated pea seed at 350 nm, which is characteristic of tryptophan. Upon boiling pea seed milk for 5 min, the fluorescence of PSFm was quenched by nearly 50%, as shown in Figure 2C, suggesting that the local tertiary structure around Trp residue was largely changed. With an increasing boiling time from 5 to 10 min, the quenching of the PSFm fluorescence became larger (Figure 2C), accompanied by a blue shift, which was indicative of a greater change in the local tertiary structure. This observation is consistent with the above CD results.

To explain the above-observed phenomenon, we analyzed the predicted structure of pea seed ferritin (Figure 2D), and found that there are four tryptophan residues located at the four-fold channels, which are surrounded by a number of hydrophobic residues. Research has demonstrated that high temperature may lead to the exposure of hydrophobic residues, which is a key factor for protein denaturation [34]. Thus, it is possible that thermal treatment causes more hydrophobic residues within the 4-fold channels exposed, producing intermolecular hydrophobic interactions, resulting in the quenching of fluorescence of PSFm.

### 3.2. Effect of Thermal Treatment on the Assembly of PSFm

The above results raise an interesting question of whether the shell-like structure of PSFm is kept. To answer this question, the assembly state of PSF was estimated by the Native-PAGE (Figure 3A), and the results demonstrated that its 24-meric shell-like structure of PSFm was kept well, although the PSFm subunit was partially destroyed by heating. To confirm this idea, TEM was used to visualize the morphology of PSFm with PSF from pea seeds as the control, and results were displayed in Figure 3. It was observed that PSF from pea seeds are distributed uniformly, with an outer diameter of 12 nm and inner diameter of 8 nm (Figure 3B). In contrast, the PSFm molecules, which were prepared by boiling for 5 min, were slightly aggregated (Figure 3C). Such aggregation behavior became more obvious with PSFm obtained by boiling for 10 min (Figure 3D). However, the shell-like structure of these two ferritin samples still maintains well, which is similar to that of PSF from pea seeds. Differently, our recent study demonstrated that, upon thermal treatment of pure PSF at 100 °C for 10 min, its shell-like structure was damaged to a large extent [24]. These findings are in good agreement with the above Native-PAGE result, demonstrating that other food components in pea seed milk have a markedly protective effect on PSFm molecules.

To further confirm this view, we also launched the DLS experiments, which may provide more information about the protein state in a solution. According to Figure 4, one population size was detected in PSF directly purified from pea seeds, which is around 6.7 nm. Similarly, the radius of PSFm obtained by 5-min boiling was almost unchanged. In contrast, upon 10-min boiling, PSFm exhibited one more population, appearing in the solution with the radius size of 98.9 nm, which was indicative of protein aggregation. These results agreed with the above TEM results, indicating that the thermal stability of PSFm is pronouncedly larger than that of pure PSF. These results again demonstrated a protective effect of other components in pea seed milk on PSFm.

### 3.3. Effect of Thermal Treatment on the Iron Content and Catalytic Activity of PSFm

Bearing in mind that natural ferritin is rich in iron, we further determined the iron content of PSFm upon thermal processing. It was found that approximately 3500 iron atoms could be sealed within one PSF shell, as determined by ICP-MS (Figure 5A). In contrast, a small amount of the stored iron within PSFm (~10%) was lost upon 5-min boiling. Differently, 10-min boiling resulted in a loss of a large amount of the total ferritin iron (about 75%). Thus, it seems that thermal treatment time is a crucial factor affecting the iron content within the ferritin cage during the preparation of pea seed milk. The shorter thermal treatment time, the higher the iron content left within the cavity of ferritin. During the pea seed milk production, thermal treatment at 100 °C for 5 min was friendly for the iron inside the ferritin nanocage under the protection of diverse food components in pea seed. To confirm this phenomenon, we also compared the morphology of iron core between ferritins from untreated pea seed and pea seed milk. Natural iron cores from PSF were homogenously distributed, as shown in Figure 5B. There was no distinct difference in the iron core distribution between PSF and 5-min boiling PSFm samples (Figure 5C), approving the above conclusion.

Besides the function of plant ferritin from legume seeds in iron storage, plant ferritin also plays a crucial role in iron detoxification [35]. This specific characteristic of ferritin not only balances the iron content inside and outside, but also helps to avoid the toxicity of ferrous iron. Similar to other plant ferritin, PSF also consists of a ferroxidase center, which is composed of Glu27, Tyr34, Glu62, His65, Glu107, and Gln141 located within the four-helix bundle of each subunit. Therefore, we continued to investigate the catalytic activity of PSFm to convert Fe(II) into Fe(III) in the presence of oxygen, aiming to clarify the relationship between iron oxidation activity and protein structure. UV absorption in the 300–330 nm spectral regions has been traditionally used to monitor the formation of Fe(III) species during the oxidative deposition of iron in the ferritins. Spectrophotometric kinetic measurements were launched to compare the catalytic activity of PSF with that of PSFm. In these experiments, Fe(II) was added into a ferritin solution with an iron-protein ratio of 200 to 1 at a temperature fixed at 25 °C. As shown in Figure 6, the UV-visible kinetic traces of untreated PSF are very smooth, indicating that no insoluble protein aggregates form during the iron oxidation process. For a 5-min boiling PSFm sample, the oxidizing rate became slightly slower with similarly smooth traces, proving that the catalytic activity of the ferroxidase center is hardly influenced. However, with an increasing thermal treatment time from 5 to 10 min, the rate of iron oxidation became very slow, suggesting that the structure of the ferroxidase center was broken. These results are in accordance with the above iron content results. It is obvious that the longer boiling time passivates the activity of ferritin as an iron-oxidizing enzyme.

### 3.4. Effect of Thermal Treatment on the Iron Supplement Activity of PSFm

Except for the detoxification activity of plant ferritin, it has been considered as a promising iron supplement in the 21st century. Therefore, we further evaluated the effect of thermal processing on the ferritin iron absorption by human intestinal Caco-2 cells. Generally, as shown in Figure 7A, PSFm samples exhibited higher iron absorption rates by Caco-2 than untreated PSF at an identical protein concentration, although the untreated PSF sample can carry more iron within its cavity. The 5-min boiling PSFm sample demonstrated the highest iron absorption activity by Caco-2, followed by 10-min boiling PSFm, and untreated PSF. There may be a few reasons responsible for this phenomenon (Figure 7B). Firstly, more iron ions and iron cores escaped from the ferritin nanocage after thermal processing and gastric digestion, which are easier to pass through the cell wall of Caco-2 cells. It has been proven that DMT1, the receptors of Fe^2+^, are distributed among the exterior surface of intestinal cells [36]. Iron cores are possibly absorbed directly by endocytosis [37]. Besides, there are still parts of PSF that maintained their entire cage structure when they go into the enteric cavity. However, according to the above results, the shell-like structure of ferritin becomes loosened after thermal treatment. It is supposed that such a loosened structure of boiled ferritin molecules may have better affinity with their receptors that are located on the exterior of Caco-2 cells. A detailed mechanism needs to be further explored in a future study. Taken together, these results support a new idea that iron from thermal-treated ferritin rather than untreated analogues can be better absorbed by Caco-2 cells.

### 3.5. Discussions

In this research, we have completed Caco-2 cell experiments to explain the absorption differences from the perspective of protein molecules. In the real food-process system, it is possible that complex components in pea seed milk will also affect the absorption of PSFm, which remains to be verified by animal experiments in the future. Additionally, there is no final conclusion about the relationship between the three absorption pathways, which can be clarified by experiments with gene-deficiency mice in the future.

## 4. Conclusions

Based on the present morphology, structure, and functionality results, the mechanism by which the structure and function of PSFm change during the pea seed milk production can be summarized. Natural ferritin is full of iron cores, which are enclosed and protected by a shell-like structure. Upon thermal treatments, the primary, secondary, and tertiary structure of PSFm was changed to some extent, as demonstrated by the CD and fluorescence spectra, resulting in a loosened structure; however, the shell-like structure of PSFm molecules was almost unchanged. Consequently, thermal processing, especially at a shorter time (5 min), did not cause PSFm to suffer a great loss of iron cores. As a result, PSFm exhibited better iron absorption rate than untreated PSF. The detailed mechanism is worth investigating in future. These results are beneficial for the exploration of plant ferritin as a promising iron supplement, providing theoretical support for food thermal processing.

## Figures and Tables

**Figure 1 foods-11-00557-f001:**
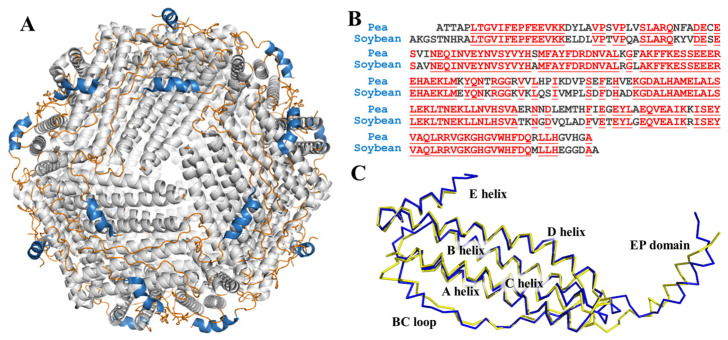
(**A**) View of structure of ferritin 24-mer down the three-fold symmetry axis: α-helix is shown in grey, loop is shown in orange, and EP domain is shown in blue; (**B**) sequence alignment of pea seed ferritin and soybean seed ferritin. Homologous residues are highlighted in red; (**C**) superposition of predicted pea ferritin subunit (blue) and soybean ferritin subunit (yellow).

**Figure 2 foods-11-00557-f002:**
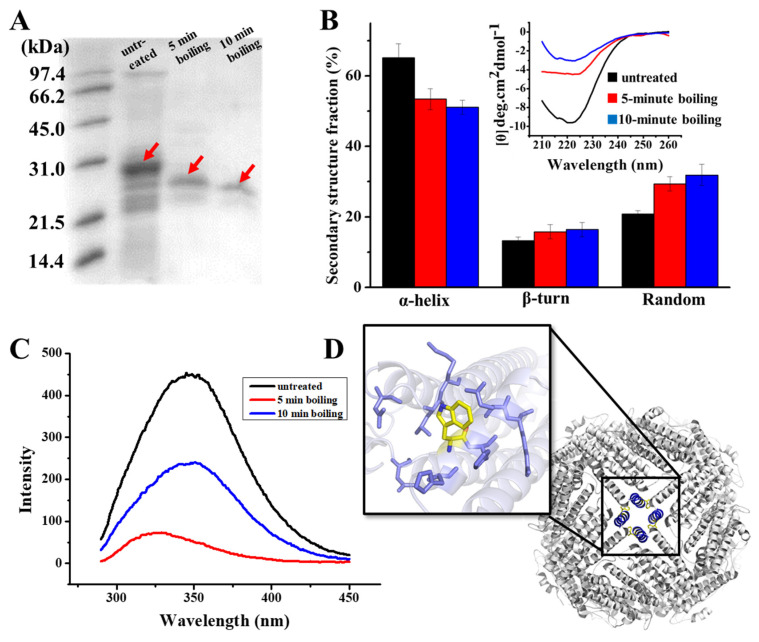
(**A**) SDS PAGE analyses of PSF purified from untreated pea seed, and two PSFm samples obtained by 5-min boiling and 10-min boiling of pea seed milk (from left to right). (**B**) CD spectra of untreated PSF and two PSFm samples, with the lines indicating the ellipticity. (**C**) Fluorescence spectra of untreated PSF and two PSFm samples. (**D**) Structural representation of tryptophan residue located at E helix located at the 4-fold channel. All measurements mentioned above were completed in three replicates.

**Figure 3 foods-11-00557-f003:**
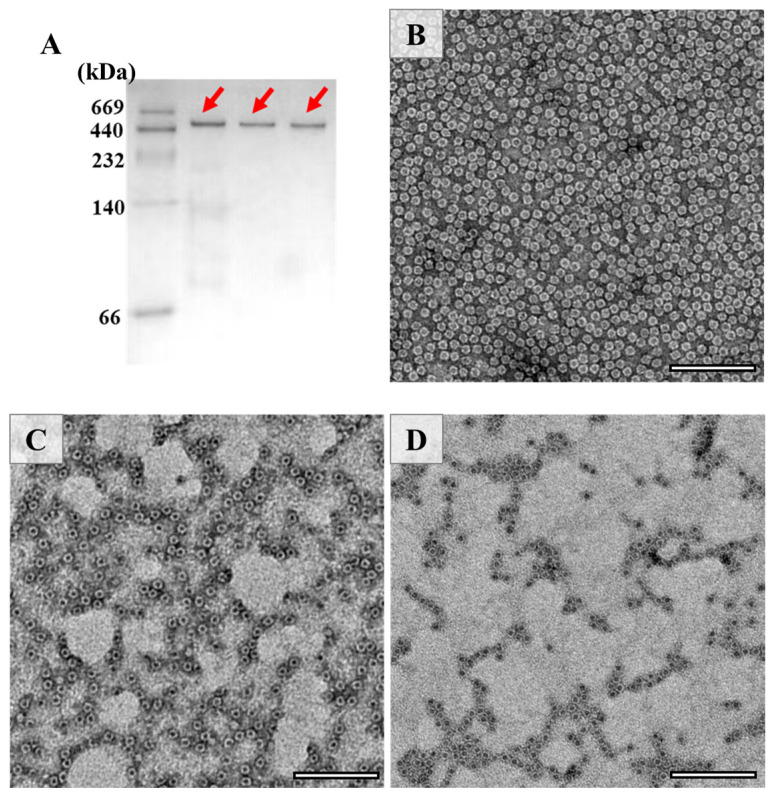
(**A**) Native PAGE analyses of PSF purified from untreated pea seed, and two PSFm samples obtained by 5-min boiling and 10-min boiling pea seed milk (from left to right). TEM images of PSF purified from pea seeds (**B**), PSFm which was obtained by boiling for 5 min (**C**), and 10 min (**D**), respectively. Scale bar represents 100 nm.

**Figure 4 foods-11-00557-f004:**
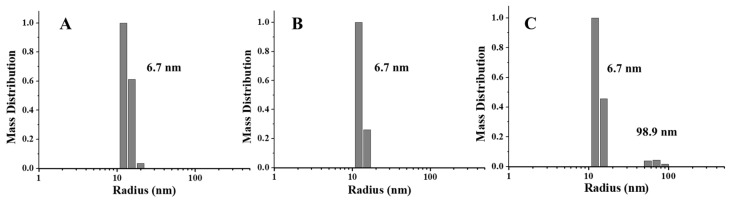
DLS analyses of PSF purified from untreated pea seeds (**A**), and two PSFm samples from 5-min boiling (**B**) and 10-min boiling (**C**) of pea seed milk, respectively.

**Figure 5 foods-11-00557-f005:**
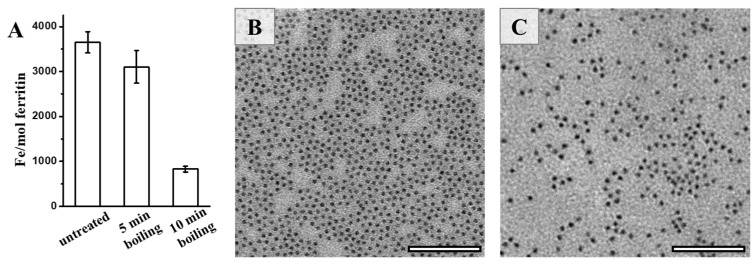
(**A**) The iron contents of untreated PSF, and two PSFm samples from 5-min boiling and 10-min boiling pea seed milk, respectively. TEM images of untreated PSF (**B**) and PSFm from 5-min boiling pea seed milk (**C**). Scale bar represents 100 nm.

**Figure 6 foods-11-00557-f006:**
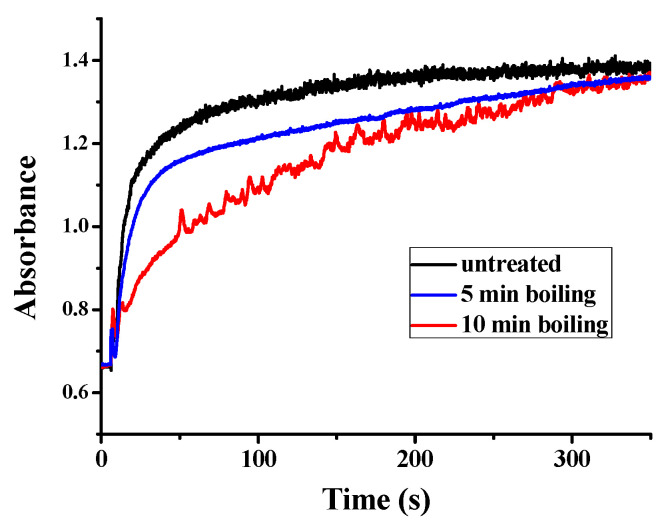
Kinetic curves of Fe^2+^ oxidation by O_2_ in the presence of untreated PSF, and two PSFm samples from 5-min boiling and 10-min boiling of pea seed milk, respectively.

**Figure 7 foods-11-00557-f007:**
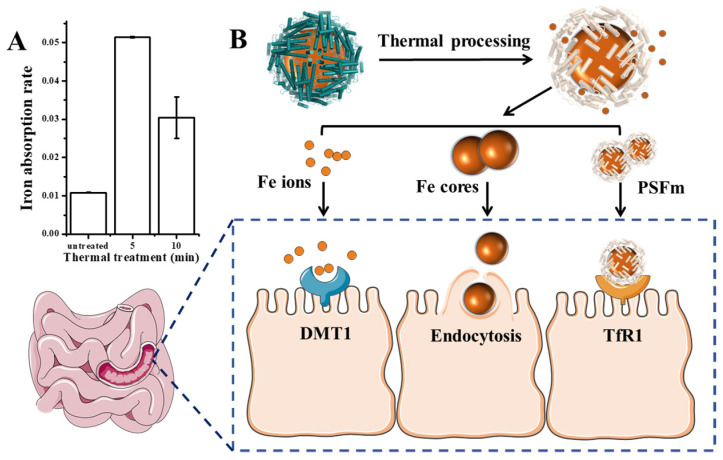
(**A**) Iron absorption rate of untreated pea seed and pea seed milk boiled for 5 and 10 min, respectively. (**B**) The possible absorption mechanism of PSFm into Caco-2 cells.

## Data Availability

Not applicable.

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
