# Peer review of "The Change in the Structure and Functionality of Ferritin during the Production of Pea Seed Milk"

_foods, 2022, doi:10.3390/foods11040557_

Round 1

Reviewer 1 Report

The authors investigated how thermal treatments (boiling for 5 or 10 minutes) influenced the structures of pea seed ferritin, as well as its iron content and ferrous oxidation activity. By using the Caco-2 cellular model, iron absorption rates from the samples were also compared. Strength of the study is that it is reasonably comprehensive, presenting not only biochemical/structural data, but also cellular evidence. A question brought about by one observation/result was also addressed in a subsequent analysis/experiment, suggesting that the analyses in the study were well-connected.

I found no major issues.

But there are a few parts where the manuscript can be made clearer/more informative. Below are some minor issues/my feedbacks for the authors’ consideration.

  1. Line 75 – “… EP domain of soybean seed ferritin makes a great contribution to its high thermo-stability.” – Can the same be said about PSF? As indicated in Lines 256-260, EP domains of PSF appeared thermal-sensitive and readily destructed by high temperature. Could the authors make a comparison/clarify this?

  1. Lines 102-103 – “2 M MgCl2 and 0.3 M sodium citrate are respectively used to adjust the supernatant, with a 1 h stirring for each step, and kept for 8 h.” – Some information seems missing here or that the statement is incomplete. Could the authors please recheck - what was being adjusted?

  1. Line 105 – For dialysis, it would be useful for others who hope to reproduce/repeat the experiment if information on membrane type and MWCO can be provided.

  1. Lines 107 & 111 – “DEAE-Cellulose column” & “Sephacryl S-300 molecular sieve column” – It would be useful for others who wish to repeat the experiment if information on column height and diameter could be provided.

  1. Line 117 – “3 Electrophoresis” – Could the authors please indicate the standardized protein amounts in the 20 uL loaded samples? Also, please change “strained” (line 124) to “stained”.

  1. Lines 131 & 150 - For dialysis, it would be useful to provide a bit more details, e.g., temperature, time/duration, membrane type and MWCO.

  1. Line 173 – “standard solution containing iron” – Was this self-prepared or a commercially available iron standard (in nitric acid) formulated for ICP-MS analysis?

  1. Lines 182 & 200 – “thermo fisher” – Meaning unclear. Please recheck. (Perhaps the authors just indicated the brand of the apparatus??)

  1. Line 197 – “5 mL of ferritin sample solution (~1.0 μM pea seed ferritin)” – For the absorption study, did the authors use pea seed milk, or pea seed ferritin purified from the latter? This needs to be indicated clearly.

  1. Line 204 – It should be “ Results & Discussion”.

  1. Lines 222-224 – “As shown in Fig. 2A, …It appears that with most of PSFm molecules still maintains its …
  • Was this (“… most of…”) based on visual comparison of the three side-by-side protein bands in Fig. 2A? If so, when comparing the “untreated” band with the other two bands, their band intensities seem to be quite different. So, I am unsure why the authors mentioned that “most of” the PSFm molecules still maintain their water solubility. Could the authors please check this part again?
  • Also, the authors could consider quantifying the intensities of the three bands in Fig. 2A, e.g., by using the ImageJ software. This may provide some semi-quantitative evidence to support their interpretation.

  1. Figure 2
  • 2A – there seems to be more other bands in the first sample lane (PSF from untreated peas seed), as compared to the other two lanes. Could the authors please clarify what could have caused this?
  • 2B – Looking at the bar chart, the difference between the 5 min and 10 min samples (red vs blue bars) seems not so clear-cut – despite the descriptions in the text (lines 249-253). Do these bars represent mean values of replicated measurements – how many replicates were done? Please indicate in the manuscript/M&M.
  • The authors could consider running a statistical test (e.g., one-way ANOVA followed by Duncan’s Multiple Range test, or any other appropriate post-hoc test) to reveal any statistical significance in the results (between the 5 min and 10 min samples).

  1. Lines 276-277 – “It is a common sense that…” - It would be appropriate to support the statement with some cited references instead.

  1. Lines 326-328 – Although the authors mentioned that there is “no distinct difference in the iron cores distribution between PSF and 5-min boiling PSFm samples”, but the two TEM images (Fig 5B vs Fig 5C) just look too drastically different. Could the authors recheck this?

  1. Section 3.4
  • If pea seed milk (rather that pea seed ferritin purified from pea seed milk) was used (please see comment #9 above), the influence of other components in the pea seed milk may have to be considered when interpreting the absorption results. Will other components in the milk promote iron absorption via any of the three routes in Fig 7B?
  • If pea seed ferritin purified from pea seed milk was used in the absorption study, could the authors clarify whether iron ions/cores released from partially/fully degraded ferritins during thermal processing will still be available for the absorption experiment?
  • Could the authors also consider the relative importance of the three iron uptake routes to the 5 min and 10 min samples? For example, is it possible that the DMTI and endocytosis routes are more crucial to iron absorption from the 10 min sample, whereas the TfR1 route might be more important for the 5 min sample?

Reviewer 2 Report

I carefully read manuscript and did not find any mistake or incorrect results and comments. I can recommend authors in further research to test more gentle treatment of pea seed milk without heating such as HPP and PEF considering that there is need to inactivate antinutrient components at the same time as uninvitable microorganisms.

Reviewer 3 Report

Manuscript title: The change in the structure and functionality of ferritin during the production of pea seed milk

Comments to authors:

Line 26: Rephrase

Line 43: use appropriate word to make sentence

Line 51: if you are talking about ferritins portion present in plants only, in this manuscript. Then no need to discuss animal ferritins here.

Line 60: Rewrite, grammatically mistake

Line 68: Need to concise and split this paragraph

Line 48: Introductory sentence are missing about pea, soybean percent of ferritin and pea seed milk, why this one need to called as pea milk seed

Line 83: Introduction needs to rewrite, in a story form, discuss your work in sequence, work story, what is specifications of your selected compound, what was the overall deficiencies, working on ferritin, iron, and protein?? Make clear sentence on which parameter you are going to check effect of thermal treatment. Why need to do this work

Line 219: boiling at which temperature?

Line 182: simplify the sentence and concise this procedure

Line 204. Where are discussions? Only results are mentioned in heading.

Line 206-210: Introductory sentence, no need in results portion

Line 218: in result portion of manuscript it’s not need to explain whole procedure, just mention the results. Concise this portion by using suitable statements regarding to results

Line 280: Reference study is missing for comparing your present study result

Line 257: Rewrite

Line 259: Ambiguous, rephrase

Line 299: also mention 5 mint boiling temperature and 10 minutes as well. If both  are same, then mention both conditions

Line 377: Results Portion needs to care full attention. Elaborate the result with reference studied and compare the results.

Round 2

Reviewer 3 Report

Great work.